# Prediction of Chronological Age in Healthy Elderly Subjects with Machine Learning from MRI Brain Segmentation and Cortical Parcellation

**DOI:** 10.3390/brainsci12050579

**Published:** 2022-04-29

**Authors:** Jaime Gómez-Ramírez, Miguel A. Fernández-Blázquez, Javier J. González-Rosa

**Affiliations:** 1Institute of Biomedical Research Cadiz (INiBICA), Universidad de Cádiz, 11003 Cádiz, Spain; javier.rosa@uca.es; 2Department of Biological and Health Psychology, Universidad Autónoma de Madrid, 28049 Madrid, Spain; miguelangel.fernandezb@uam.es

**Keywords:** aging, MRI, machine learning, XGBoost, feature importance, shapley values, brain segmentation, cortical parcellation, age prediction, biological aging

## Abstract

Normal aging is associated with changes in volumetric indices of brain atrophy. A quantitative understanding of age-related brain changes can shed light on successful aging. To investigate the effect of age on global and regional brain volumes and cortical thickness, 3514 magnetic resonance imaging scans were analyzed using automated brain segmentation and parcellation methods in elderly healthy individuals (69–88 years of age). The machine learning algorithm extreme gradient boosting (XGBoost) achieved a mean absolute error of 2 years in predicting the age of new subjects. Feature importance analysis showed that the brain-to-intracranial-volume ratio is the most important feature in predicting age, followed by the hippocampi volumes. The cortical thickness in temporal and parietal lobes showed a superior predictive value than frontal and occipital lobes. Insights from this approach that integrate model prediction and interpretation may help to shorten the current explanatory gap between chronological age and biological brain age.

## 1. Introduction

Magnetic resonance imaging (MRI) has revolutionized clinical neuroscience, assisting in better diagnostics and playing a crucial role in helping to close the gap between basic and clinical research [1]. The contrast and detail that MRI achieves are unparalleled in detecting brain abnormalities, tumors, and micro-hemorrhages. Technological advancements—in particular, software development—are making it possible to assess conditions that went previously undetected [2]. Furthermore, faster imaging is helping to alleviate the clinical burden in patients and clinicians alike, with AI-based analysis drastically reducing the time required for image reconstruction [3,4]. In addition, the ever-increasing computational capacity and availability of AI techniques are accelerating the use of automated image post-processing for diagnostics and prognosis assessment in clinics and hospitals. Volumetric analysis aiming at quantifying the volume of brain structures and the thickness and gyrification of cortical areas can be particularly effective in flagging brain abnormalities in large datasets [5,6].

MRI is also helping to characterize the neuroanatomy of healthy brain aging across ages and conditions. Several studies show the intricacies of morphological changes visible for the whole brain [7,8], as well as for the cerebral cortex [9], subcortical grey matter structures, and white matter integrity differences [10].

Changes in functional abilities and brain structural alterations, such as atrophy, are expected during aging. However, there is not always a clear separation line that distinguishes the effects of brain aging from neurodegeneration. For example, white matter hyperintensities or microbleeds are present in neurological conditions such as Alzheimer’s disease [11], and they have also been found in aging asymptomatic individuals [12]. Both normal aging and neurodegenerative diseases are accompanied by brain morphological changes, notably atrophy or the loss of tissue volume resulting from cellular death. At advanced stages of the disease, the cellular loss or synaptic pruning associated with atrophy is more easily recognizable than at early stages, making the early detection of neurodegenerative pathologies particularly challenging.

Whole-brain atrophy intended as a diminution of brain volume normalized to the intracranial volume can be measured with T1-weighted structural MR images. Both grey matter and white matter decline with aging, with, at the same time, an increase in the size of the ventricles and cerebrospinal fluid (CSF) volume [13].

Among the subcortical structures in the limbic system and basal ganglia, the hippocampus is arguably the best characterized anatomic volume concerning aging. Hippocampal volume loss is also a well-recognized biomarker for the diagnosis of Alzheimer’s disease in cross-sectional [14], longitudinal [15], and meta-analysis [16]. The variation in hippocampal volume as a function of age studied in the UK Biobank Imaging dataset (*N* = 19,793) shows an acceleration in hippocampal volume reduction in middle age (starting at around 50 years of age) [17]. Age-related atrophy in other subcortical structures varies, with a more rapid decline in the thalamus and putamen compared to caudate and amygdala [18]. However, the software libraries used for automated segmentation can introduce variability in the volume estimates [19].

The “last in first out” hypothesis states that the last brain regions to develop tend to be the first ones to decline. The hypothesis is rooted in the idea of the lifespan composed of two stages: first, a developmental phase, and then followed by an aging phase characterized by progressive cortical thinning, with the onset linked to late-maturing regions of the brain, such as the heteromodal association cortices [20]. Cortical thinning spans widespread cortical regions with unequal effects, e.g., being more prominent in the prefrontal cortex and less so in the parahippocampal cortex [9,21]. Longitudinal estimates of cortical thinning converge with cross-sectional studies that find significant thinning in the heteromodal association cortex with a later expected atrophy in the primary cortex, in line with the “last in first out” hypothesis [22].

Since, as already mentioned, brain volume decline is associated with age, it is possible, at least in principle, to use the volumetric measurement of brain atrophy to estimate its age. It is, however, useful to distinguish between biological and chronological age. The gap between both ages could indicate the aging pace of a given subject; that is to say, a subject with biological age larger than chronological age would imply a faster than expected aging decline. On the other hand, a person more chronologically than biologically aged could indicate that that person is aging more slowly than expected.

Neuroimaging-based studies for chronological age prediction use different features, acquisition techniques, and MRI sequences [23,24,25]. In [26], a supervised regression model used to predict age using the Destrieux atlas of cortical parcellation, obtained a mean absolute error of 4.05 years [27]. Cortical thickness combined with diffusion MRI has also been used as an input for predicting brain age in multiple regression models [28,29]. Studies comparing the performance between linear and nonlinear modeling approaches (e.g., neural networks) and combining a wide range of features have shown that performance gain from larger training can be limited if data are obtained with different acquisition protocols [30].

“Brain age” is used in the machine learning literature as the age estimate in a regression model [31]. Accordingly, the delta or difference between the brain age (estimated) and the chronological age (given) would be indicative of the health status of the subject. However, for this approach to be sensible, it requires a model producing accurate estimates of the biological age. In [32], a bias-adjustment technique was shown to shorten the delta or difference between the chronological and estimated brain age. However, the pertinence of any approach that tries to reduce the above-mentioned delta will need to be confronted with direct assessments of brain age. Recently, new cellular and molecular approaches for brain senescence and decline have been put forward. For instance, transcriptome profiling [33], DNA methylation [34,35,36], or immune metrics such as the inflammatory clock of aging (iAge) [37], aim to capture cellular senescence, holding promise for understanding longevity and neurodegenerative processes.

Although the association between the thinning of cortical areas and aging is recognized, how cortical thinning and overall brain atrophy progress in elderly healthy subjects is an open problem. In this study, we address this issue by building a machine learning model to predict chronological age in a large set of subjects using brain segmentation and cortical parcellation data, collected in a time horizon of 6 years. Different from other studies, we tackled not only the model prediction capabilities but also the model interpretability to assess the importance of the different brain areas, including whole brain and subcortical volumes and the cortical thickness of sulcal and gyral areas for age prediction using SHAP (Shapley additive explanations) [38].

## 2. Methods

### 2.1. Study Participants

The dataset used in this study comes from a single-center, observational cohort study [19,39,40]. The participants are home-dwelling elderly volunteers, 69–88 years of age, and without relevant psychiatric, neurological, or systemic disorders. The participants signed informed consent and undertook a yearly systematic clinical assessment, including medical history, neurological and neuropsychological examinations, and brain MRI. Apolipoprotein E (APOE) genotype was also studied, with total DNA isolated from peripheral blood following standard procedures. Ethical approval (CEI PEI 46_2011-v2014) was granted by the Research Ethics Committee of Instituto de Salud Carlos III, and written informed consent was obtained from all of the participants. The authors assert that all procedures contributing to this work comply with the ethical standards of the relevant national and institutional committees on human experimentation, and with the Helsinki Declaration of 1975 and its later amendments.

Of the initial 1213 subjects, those diagnosed with mild cognitive impairment (MCI) or dementia were excluded, resulting in a cohort of 948 healthy elderly subjects. Cognitive status was determined with the Mini-Mental Status Examination (MMSE), free and cued selective reminding test (FCSRT), semantic fluency, 101 digit-symbol test and Functional Activities Questionnaire (FAQ).

The subjects were assessed yearly for six years, with the number of yearly visits per subject varying from 1 to 6 visits. The distribution of the number of visits with the corresponding neuroradiological and cognitive assessment is as follows: 19.2% of subjects with one initial assessment completed, 11.3% of subjects with 2 assessments completed, 8.5% with 3 assessments completed, 8.9% of subjects with 4 assessments completed, 23% of subjects with 5 assessments, and 29.1% of subjects with all 6 assessments completed.

### 2.2. MRI Data Acquisition and Preprocessing

For all of the subjects, brain MRIs were collected in 6 yearly visits: 948 in the first visit, 744 in the second, 711 in the third, 622 in the fourth, 529 in the fifth, and 364 in the sixth visit.

The imaging data were acquired on a 3T General Electric scanner (GE Milwaukee) utilizing the following configuration: T1-weighted inversion recovery, flip angle 12${}^{\circ }$, 3-D pulse sequence, echo time Min. full, time inversion 600 ms, receiver bandwidth 19.23 kHz, field of view = 24.0 cm, slice thickness 1 mm, Freq. × Phase (288 × 288).

The preprocessing of MRI 3 Tesla images consisted of generating an isotropic brain image with non-brain tissue removed [41]. We used the FreeSurfer cortical surface reconstruction pipeline as the initial preprocessing step. The postprocessing was performed with FreeSurfer [42], version freesurfer-darwin-OSX-ElCapitan-dev-20190328-6241d26 running under Mac OS X, product version 10.14.5 as described in [43]. Parcellation was performed using the Destrieux cortical atlas, which is based on the division of the cortex into gyri or sections of the cortex visible on the pial view, and sulci or the hidden parts of the cortex according to the curvature value of the surface [27]. Figure 1 shows two views of the Destrieux atlas parcellation of the left hemisphere.

### 2.3. Anomaly Detection with the Isolation Forest Algorithm

In this study we computed the brain volume, the mean subcortical volumes of seven structures—thalamus, putamen, hippocampus, caudate, pallidum, amygdala, accumbens—and the mean cortical thickness of the 148 regions of interest defined in the Destrieux atlas. In the first stage, images were manually assessed for quality, and scans considered not suitable for analysis due to artifacts were discarded. In the second step, we performed anomaly detection using the scikit-learn Isolation Forest implementation [44], resulting in a total of 3514 MRI examinations. The alogrithm’s parameters—number of estimators, number of samples, and contamination level—were set to the default values.

Isolation Forest is an ensemble method [45] used as an outlier detector in a variety of datasets [43,46,47]. The algorithm works by isolating each point in the dataset to assess whether the point is an outlier. The idea behind the algorithm is that points that are outliers are easier to separate than those that they are not. Thus, for each point and each feature, a range between the minimum-maximum values is declared. The algorithm randomly selects a feature and a value and, depending on where the value falls in the range, the range is switched upwards to the maximum or downwards to the minimum value. The algorithm proceeds iteratively until the point is isolated; that is, the point is alone inside the range for all features. Finally, the algorithm considers a point as an outlier depending on the number of iterations required to isolate it.

### 2.4. Statistical Analysis

The sample size of the study (number of MRI scans) and demographic (age, sex, educational attainment level) and genetic information (presence of the APOE4 allele) of each subject in the study is summarized in Table 1. The MRI scans were performed in a time span of 6 years. The number of MRI examinations that passed visual inspection and automatic anomaly detection is as follows: year 1: 857 scans, year 2: 683 scans, year 3: 658 scans, year 4: 555 scans, year 5: 450 scans, year 6: 311 scans, for a total of 3514 MRI examinations.

Table 2 shows the results of the segmentation and cortical parcellation analysis. The brain-to-intracranial-volume ratio (Brain2ICV) in % or brain-to-intracranial-volume ratio and the estimated volume of seven subcortical structures for both hemispheres in mm${}^{3}$ are followed by the average cortical thickness in mm of the regions defined in the surface-based atlas known as Destriaux atlas [27].

The total intracranial volume (eTIV) estimated by FreeSurfer has previously been reported to have linear correlations of 0.9 with manually estimated intracranial volume [48,49]. Depending on whether CSF is included or not, one can dissociate the total brain volume (TBV) from the total intracranial volume. The normalized TBV is widely used as an index for brain atrophy, as the head size remains stable across the life span and serves as a good measure to reduce between-subject differences with regard to maximum brain size. Whole-brain volume, on the other hand, changes throughout the life span of an individual [50,51]. Measurements of total brain volume (TBV) with FreeSurfer are robust across field strength [52]. The total intracranial volume acts as a scaffolding of the brain and sets an upper bound for the brain’s volume. Accordingly, it is possible to build a proxy of the brain atrophy that an elder person went through in their adult life by computing the ratio between the brain volume (TBV) and the total intracranial volume (eTIV), which represents the upper limit of brain volume [53]. Thus, $Brain2ICV=\frac{TBV}{eTIV}$.

The relationship between volumetric and thickness estimates and the variables Sex and APOE4 is investigated using regression analysis. We conduct hypothesis testing on the regression coefficients obtained in the regression model $Y=\beta_{0}+\beta_{1}X_{Sex}$, where the target variable Y is the chronological age and $X_{Sex}$ is the sex of the participants.

The effect of the APOE4 allele on volumetric or cortical thickness estimates in asymptomatic individuals is approached using ANOVA analysis with the three-valued variable APOE4.

For the sake of illustration, Figure 2 shows the intracranial volume segmentation and cortical parcellation analysis obtained for four subjects in the study. The summary of the automated segmentation and cortical parcellation of the 3514 scans included in the study are shown in Table 2.

### 2.5. Age Prediction Analysis

We aim to predict the chronological age of elderly healthy adults using structural MRI volumetric analysis and demographic features of interest, such as sex, educational attainment level, and the genetic risk factor in dementia (APOE4). We built two types of predicting models, linear and nonlinear, and we assessed their performance by testing the model predictions on a held-out dataset of points not used for training.

Schematically, the prediction problem we aim to resolve can be succinctly described as the supervised learning model shown in Equation (Equation 1)
(1)Γ(X)→Y
such that the function Γ maps the input space *X = (sex, educational attainment level, APOE4), (subcortical volume estimates), (cortical thickness estimates)* into the output space *Y = (chronological age)*.

The dimensionality of the input space X of the model is 153, including the sex of the individuals (female or male), education level, and the apolipoprotein E gene called APOE4, together with 150 brain imaging features, namely the brain-to-intracranial-volume ratio (Brain2ICV), the volume estimates of seven subcortical structures (caudate, pallidum, putamen, thalamus, hippocampus, amygdala, and accumbens), and the cortical sulci and gyri thickness based on the parcellation defined in the Destrieux cortical atlas (Figure 1).

We built two supervised learning models: partial least squares (PLS) and extreme gradient boosting (XGBoost).

The PLS regression model extracts the components of the input X and the output Y that explain the most shared variance among X and Y [54]. Thus, the two components maximally correlated correspond to the first component in feature space X and the target Y. PLS is based on principal component analysis, so it deals with multi-colinearity in the input space by finding a linear transformation W such that the new input space is linearly independent while, at the same time, maximizing the covariance between Y and X. Thus, *X* is transformed into $X^{\prime }$ by a linear transformation *W* as in $X^{\prime }=XW$.

The nonlinear supervised learning model of choice is the decision-tree-based ensemble machine learning algorithm called extreme gradient boosting, or XGBoost for short. XGBoost [55] is an implementation of a gradient-boosted trees algorithm that uses a regularized gradient-boosting algorithm to accurately predict the target variable: in our case, the chronological age. XGBoost combines weaker models with stronger ones, which are added to improve the overall performance. During training, the gradient descent algorithm minimizes the loss by adding new trees to predict more accurately than weaker decision trees. The models here are all regression trees mapping the input data points (X) to one of the tree’s leaves, which contains the age (Y) that we want to predict. The objective function that the algorithm tries to minimize combines the difference between the predicted and target outputs and a penalty term for model complexity.

Formally, given the dataset tuple (X,Y), the gradient tree boosting algorithm tries to minimize the differentiable loss function *L* (Equation (Equation 2))
(2)L(Y,Γ(X))
where Γ(X) represents an ensemble of n regression trees that are sequentially added to incrementally better predict the residuals of previous trees. Formally,
(3)$$\Gamma_{n}\left(X\right)=\Gamma_{n-1}\left(X\right)+\alpha_{n}g_{n}\left(X,\rho_{n-1}\right)$$
with $\alpha_{i}$ and $\rho_{i}$i=1..n denoting the regularization and the residual parameter, respectively, for the $i_{th}$ tree. The function $g_{n}$ is trained to predict the residuals of the precedent tree in the forest ($\rho_{i-1}$).

Feature importance analysis provides insight into the workings of the predictive model used, allowing for the interpretability of the results. Shapley additive explanations, or SHAP for short, was originally developed in cooperative or coalitional game theory and in contrast to non-cooperative Nash equilibrium models [56]. Shapley values show how much a given feature changes the prediction compared to the prediction at the baseline value of that feature. The Shapley value [38]-based approach is being increasingly used by the machine learning community to deal with the interpretable feature subset selection problem [57].

Formally, the Shapley value Φ is defined via a value function ν of all features in a set S. Specifically, the Shapley value of a feature value is its contribution to the payout (e.g., if the average prediction for all instances is 0.9 and the actual prediction is 0.8, the payout is 0.1) weighted and summed over all possible value combinations.
(4)$$\Phi_{j}\left(\nu \right)=\sum_{S\in \left\{x_{1},...,x_{p}\right\}\setminus \left\{x_{j}\right\}}\frac{|S|!(p-|S|-1)!}{p!}\left(\nu \left(S\cup \left\{x_{j}\right\}\right)-\nu \left(S\right)\right)$$
where *p* is the number of features, *S* is the subset of features, *x* is the vector of feature values of the particular instance to be explained, and ν(S) is the prediction for feature values in *S*, marginalized over features that are not included in the set *S*.

The Shapley value is arguably the best permutation-based method for explaining the effects of feature values in the average prediction. An important drawback of Shapley values is that they provide additive contributions (attributions) of explanatory variables. If the model is not additive, then the Shapley values may be misleading. For a more in detail description of SHAP values, see [39] and the references within.

## 3. Results

We conducted hypothesis testing to investigate the relationship between volumetric and thickness estimates and the variables Sex and APOE4.

In the first case, Sex∼Brain volumetric and thickness, the null hypothesis is $H0:\beta_{1}=0$, and the alternative hypothesis, $HA:\beta_{1}\neq 0$. Using the *p*-value method, H0 is rejected when the *p*-value of the test statistic is small (e.g., less than 0.05(*) or 0.01(**)). The null hypothesis is rejected for Sex in the Brain2ICV variable, all of the subcortical structures, and in the large majority of cortical thickness areas (Table 2).

Carriage of the APOE4 genotype is the main genetic risk factor for developing late-onset Alzheimer’s disease. The relationship between APOE4 and brain volumetric and thickness estimates is investigated using ANOVA, the hypotheses of interest being: $H0:\mu_{1}=\mu_{2}=\mu_{3}$ and H1: means are not all equal. The results of the ANOVA analysis with the three-valued variable APOE4 is indicated in Table 2, column “APOE PR(>F)”. ANOVA analysis of the APOE4 allele fails to reject the null hypothesis in the large majority of subcortical and cortical regions. These results are not surprising since the volume and thickness were not adjusted for cerebral volume, and effect sizes for sex differences are to be expected in either volume estimates and cortical thicknesses [58,59].

Our results are in agreement with recent studies showing no overall risk effects associated with APOE4 in healthy adults for cortical thickness and subcortical volume [60]. Nevertheless, we find that the APOE4 genotype may have a deleterious effect on hippocampal and amygdala volumes, which is in agreement with the literature on atrophy of the hippocampus and amygdala in healthy elderly and impaired memory individuals [61,62].

Table 3 shows the results for the age predictors built, both linear and nonlinear. The performance is evaluated for the holdout set. The dataset is split into train and test sets. We use 75% of data for training and the remaining 25% for testing the model performance on unseen data.

For the partial least squares regression model (PLS), the age of the subjects can be estimated with a maximum residual error (MAE) of 2.57 years. Other metrics, such as the maximum residual error (MXE), the mean absolute percentage error (MAPE), and the median absolute error (MEDAE), are also shown in Table 3 (first row).

Next, we build, train, and tune using cross validation the nonlinear regressor using the extreme gradient boosting algorithm (XGBoost). The optimization or tuning of the hyperparameters -*η, γ, subsample, colsample by tree, max depth, and min child weight* is performed using the grid search method [63].

The hyperparameters η and γ correspond to the learning rate and the minimum loss reduction, respectively. The learning rate, also called the shrinkage rate, η, is used to prevent overfitting. The minimum loss reduction γ acts as a pseudo-regularization hyperparameter in gradient boosting: the higher the γ, the higher the regularization. The hyperparameters *subsample* and *colsample by tree* control the sampling of the dataset at each boosting round: *subsample* is the fraction of observations (rows) and *colsample by tree* is the fraction of features (columns) used to train each tree. The hyperparameters *max depth, min child weight* add constraints on the architecture of the trees: *max depth* is the maximum number of nodes allowed from the root to the farthest leaf (a very large max depth can cause overfitting) and *min child weight* is the minimum weight required in order to create a new node in the tree.

The XGBoost model achieves a mean absolute error (MAE) equals to 2.03, which is a 21% improvement relative to the partial least squares (PLS) model (MAE=2.57) as shown in Table 3 (last row).

For the sake of comparison, the performance metrics of the two models built—PLS and XGBoost—are compared with two dummy models: one always tries to predict the mean and the other the median age of the subjects. The mean absolute error (MAE) achieved by the former is 3.258 (PLS MAE = 2.57, XGboost MAE = 2.030) and the median absolute error (MEDAE) achieved by the latter is 2.850 (PLS MEDAE = 2.158, XGboost MEDAE = 1.745) (Figure 3). The $R^{2}$ by the dummy models is, as expected, 0.0, whereas $R^{2}$ = 0.353 for PLS and $R^{2}$ = 0.5915 for XGBoost.

Our results show that both linear and nonlinear models achieved a maximum mean absolute error of fewer than three years, which is below previous studies. For example, in [26], the mean absolute error achieved using a Gaussian process regression (GPR) algorithm in a sample of 2911 cognitively normal subjects (age 45–91 years) was 4.05. Relevance vector machine algorithms have been used as well to predict brain age, as in [28,64,65], all with a mean absolute error of around 4.5 years. Neuroimaging techniques other than T1-weighted MRI for brain age prediction—for example, diffusion tensor imaging in a cohort of 188 subjects aged 4–85 years—obtained a mean absolute error, depending on the age group and sex, between 6 and 10 years [66]. Neuroanatomical prediction studies of biological maturity using resting-state fMRI [67] in a cohort of 238 young subjects 7–30 years reported 55% of the sample variance, and, in multimodal studies combining MRI and diffusion-weighted imaging to predict child age, nonlinear modeling was able to account for more than 92% of the variance in age [68].

### 3.1. Feature Importance

In addition to build predictive models of chronological age from age, sex, APOE4 gene, and brain volumetric estimates, it is of particular interest to understand the relative feature importance among predictors. The identification of brain regions and structures may help us to characterize the localized effects of normal brain aging and to use the age predictors as potential biomarkers for neurodegenerative diseases [69,70].

The approach used here for addressing the feature selection problem at hand is Shapley values. Shapley or SHAP values can be used to decompose predictions into the sum of the effect of each feature.

Figure 4 shows the twenty most important features for predicting the chronological age of the subjects in the test set according to the Shapley value method. The most important features are the brain-to-intracranial-volume rate (Brain2ICV) followed by the volume of the hippocampi and the sulcal thickness of the occipito-temporal medial and lingual cortical regions.

Figure 4 shows the relative importance of the features for predicting the chronological age. On the left, Figure 4b, are the absolute Shapley values of the most important features, and on the right, Figure 4b, are the SHAP values for each subject. Thus, Figure 4a is the aggregate plot of Figure 4b. Every dot in Figure 4b represents a different subject in the test set, colored by red or blue if the feature for that subject tends to push towards the right (more age) or to the left (less age) in the model predictions. For example, large (red) values in the top feature (Brain2ICV) decrease the prediction of age; that is, subjects with a larger brain-to-intracranial-volume ratio will tend to be less aged than subjects with small (blue) values. The same can be said for the rest of the features according to the figure, with, however, a few exceptions, such as the thickness of the rectus in the left hemisphere, the thickness of the sulci in the circular anterior insula, and the left caudate volume, which seem to have the opposite effect (big values seem to push towards more age).

The brain-to-total-intracranial-volume ratio or $Brain2ICV=\frac{TBV}{ICV}$ and the hippocampi are, according to SHAP values, the most important variables in predicting chronological age. The key role of the hippocampus in learning and memory makes it a structure of particular interest to study the effects of normal aging in the brain [71,72]. A recent study with a large normative database confirms that hippocampal volume loss accelerates in middle age [17]. Interestingly, the brain-to-intracranial-volume ratio and the hippocampi volume are more important (SHAP value) in predicting chronological age than the cortical thickness of individual regions. We argue that the thickness of cortical areas could be more sensitive to the biological age than it is to the chronological age. How the current results are subject to change when predicting the biological age of the brain using, for example, DNA methylation directly from the brain [35] or the inflammatory aging clock [37], is a matter of future studies [73].

### 3.2. Feature Importance of Cortical Gyri and Sulci

The evaluation of the importance of features for predicting chronological age can also be conducted by grouping the cortical areas based on their location (hemisphere and lobe) and type (sulci or gyri). Since the Destriaux atlas used for automated cortical parcellation identifies the location of sulci and gyri in the different brain lobes and the insula, it is possible to plot the relative importance of the cortical areas for predicting chronological age.

Figure 5 depicts the aggregate importance of sulci and gyri cortical areas according to hemispheres Figure 5a and lobes Figure 5b. The aggregate importance is computed as the mean average of the SHAP values normalized. According to the SHAP values, the total sulcus thickness in the right hemisphere contains, on average, more information regarding the chronological age of the subject than gyrus thickness in either hemisphere (0.238, 0.248) and sulcus thickness in the left hemisphere (0.169). Our results indicate that the thickness of temporal lobe areas is a better predictor of chronological age than the frontal lobe, and, particularly, in cortical sulcal areas. This is in agreement with neuroanatomical evidence that indicates that between one-half to two-thirds of the cortical surface lies in the sulci and the lateral fossa of the brain [27].

Figure 5b shows the relative importance according to the SHAP values of sulci and gyri in the different brain lobes and the insula. The sulci areas in the temporal lobe are more important for predicting age than the rest of the parcellations. It is important to realize that the estimates depicted in the figure must be interpreted only in relative terms and not in absolute terms; that is to say, the sulci in the temporal lobe are more important for predicting chronological age than sulci and gyri in the frontal lobe. Likewise, the total thickness of sulci and gyri in both frontal and parietal lobes contains, according to this analysis, less information regarding the chronological age than the thickness of sulci and gyri in the temporal lobe.

## 4. Discussion

It has been suggested that the estimates of brain age from neuroanatomical data may suffer from systematic bias manifested as underestimated brain age for older subjects and overestimated for younger ones [74]. A plausible explanation for this bias may rely on the age distribution with non-normal or skew-normal distributions playing a putative role in introducing a confounding effect [75]. Nevertheless, characteristics of the sample data used to train the models can lead to different estimates of the mean absolute error in age predictions [76], with some studies having a wide range of ages [66,77,78] while others concentrating on a limited range [68].

In a healthy adult population, we may expect both maturation and aging effects in the brain, with an opposing rate of incidence; that is, being maturation prevalent during youth and aging during old age. The cortical thickness declines due to normal aging, with a more visible cortical thinning effect in areas responsible for executive processing tasks and episodic memory retrieval, which are also known to be associated with age-related cognitive decline [79].

The last in, first out hypothesis implies that late-maturing regions of the brain, such as the heteromodal association cortices in the frontal lobe, could be particularly vulnerable to the age-related loss of structural integrity [20]. However, the idea of brain aging following a simple pattern of neural decline with the prefrontal cortex as the region most disrupted is being challenged by studies that suggest a more intricate picture. For example, the prefrontal cortex could play a compensatory role [80] or, as suggested in [81], normal aging could reflect non-specific neural responses rather than the predictable decline of target areas or compensation.

While this study benefits from being single-center using an identical protocol for image acquisition, the generalization of prediction models coming from multi-site datasets is becoming a crucial problem in need of a solution [82]. A promising approach for achieving acceptable harmonization and coherence prediction results is transfer learning [83]. A recent study [84] used a transfer learning approach to build brain age prediction models from diffusion MRI datasets extracted from the CamCAN repository [85] achieved MAE within 4 to 5 years. Although not directly comparable, it is worth noting that the range of chronological ages in our study is within 69-88 years, while in [84] prediction is from an adult population (18 and older) with a smaller dataset (616 samples).

For a comprehensive evaluation of machine learning algorithms for age prediction, see [86]. Support vector regression (SVR) achieved the best accuracy in predicting age in healthy subjects: MAE∼3 and MAE∼5 years for training and test sets (N=788+88). The performance of the SVR in our dataset is shown in the Appendix A. The reduction in the prediction error indicated in our study may respond to several factors, such as the large single-center dataset, the scalability of the tree boosting algorithm, and the mapping from input features to output data. The input space in this study is the product of segmentation and parcellation methods used to extract subcortical volumes and cortical thickness, which is admittedly more computationally demanding than grey matter intensities extracted from the whole brain [87,88].

Lastly, it should be remarked that our model predicts chronological age and not biological brain age, which cannot be directly measured but could, however, be estimated via proxies such as DNA damage.

## 5. Conclusions

We trained machine learning models to predict chronological age. We find that the non-linear regression modeling extreme gradient boosting (XGBoost) obtains better results than the partial least squares (PLS) model; in particular, XGBoost achieves a mean absolute error of 2 years. Secondly, we find that the best predictor of chronological age is the brain-to-intracranial-volume ratio, followed by the hippocampi volume. Thirdly, the thickness of sulci is more important in predicting age than the thickness of gyri, and this is particularly the case for sulci in the temporal lobe. Our results show that simple volumetric features, such as the brain-to-intracranial-volume ratio and hippocampal volume, are no less important in predicting chronological age than the cortical thickness of any specific area in the Destriaux atlas. Ultimately, these results enable future research in the gap between the brain’s biological and chronological age. The operationalization of this gap using the methodology proposed here may derive into a frailty index for healthy individuals or a potential biomarker for neurodegenerative disorders.

## Figures and Tables

**Figure 1 brainsci-12-00579-f001:**
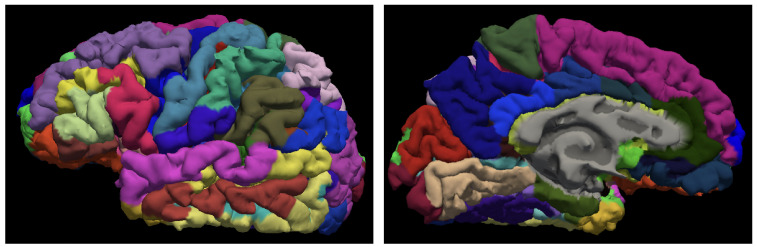
Cortical parcellation of the left hemisphere according to the Destriaux atlas.

**Figure 2 brainsci-12-00579-f002:**
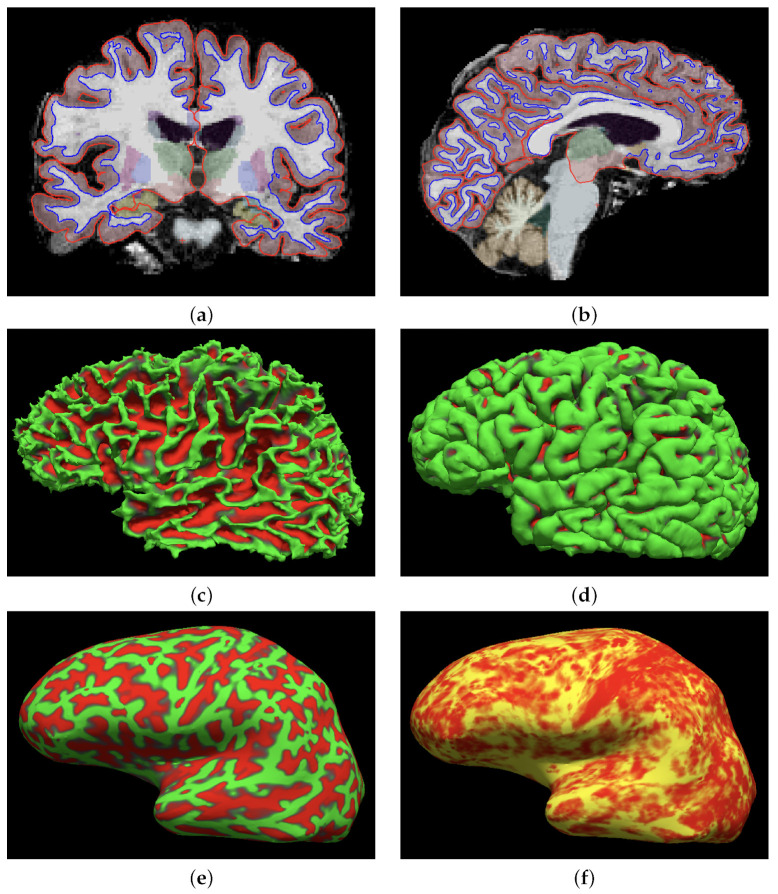
(**a**,**b**) show the coronal and sagittal segmentation results. The edge color blue indicates the demarcation of the white matter surface, and the red edge the pial surface. Plots (**c**–**f**) show the three-dimensional view of the surface analysis for the same subject. (**a**) Coronal view. (**b**) Sagittal view. (**c**) 3D view of white matter surface. (**d**) 3D view of pial surface. (**e**) 3D view of the inflated surface: giry (green) and sulci (red). (**f**) 3D view of the thickness map of the inflated surface.

**Figure 3 brainsci-12-00579-f003:**
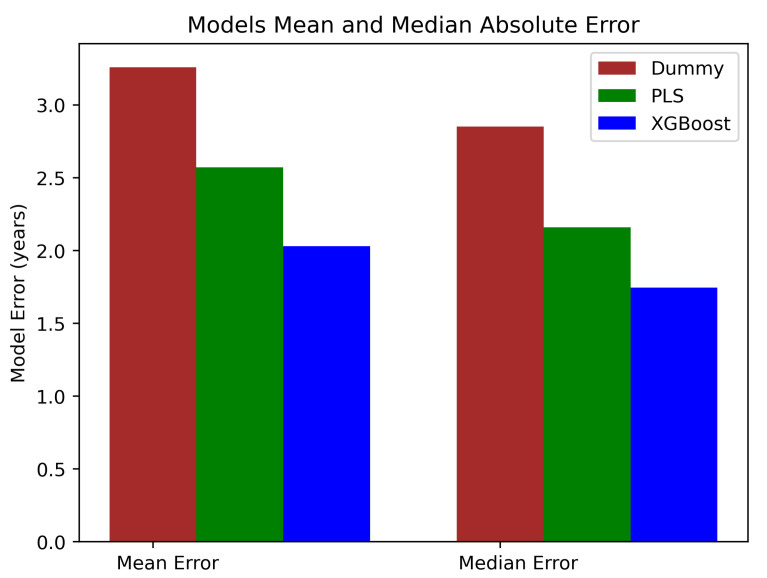
Mean and median absolute error for the dummy, PLS, and XGBoost models. The XGBoost model shows superior results to the PLS.

**Figure 4 brainsci-12-00579-f004:**
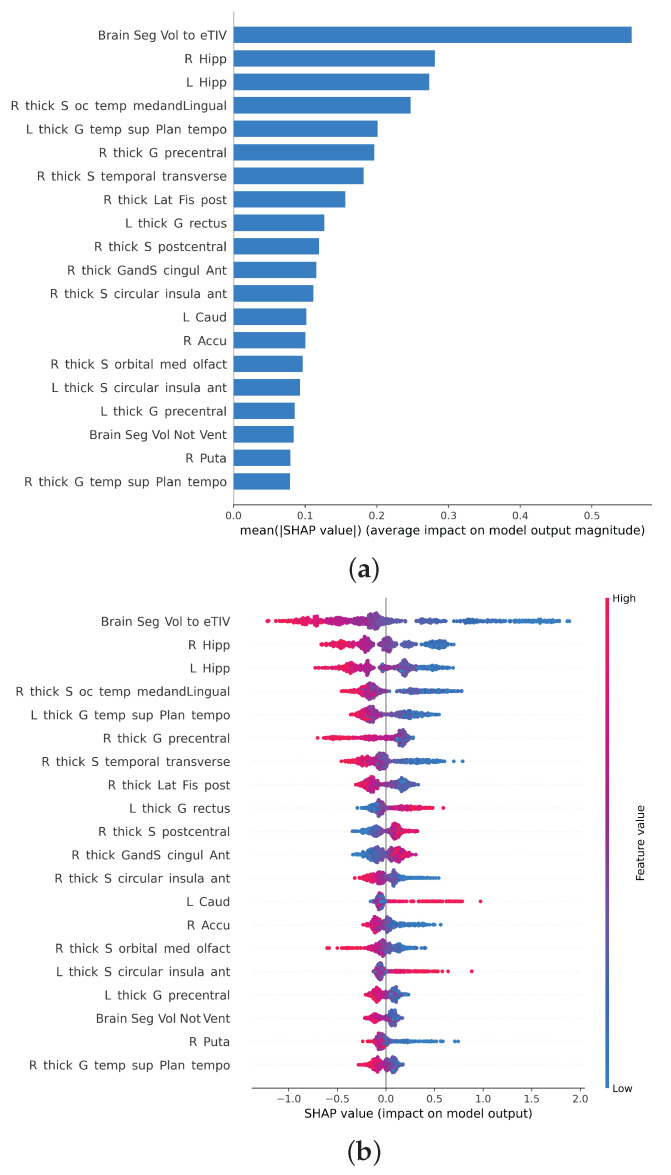
Study of the most important features based on the computation of the SHAP values for each feature and sample of the total of 804 subjects in the test set, both in aggregate (**a**) and for each data point (**b**). The vertical axis of each figure represents the features ranked by importance (top to bottom) calculated as the sum of the SHAP value magnitudes over all samples (horizontal axis). (**a**) Shapley values averaged for all subjects. The most important feature according to the SHAP values is the brain-to-intracranial-volume rate, followed by the volume of the hippocampi. (**b**) Shapley values for each subject. When the point distribution is clustered around 0, it indicates that the feature is unimportant; the more spread the distribution is, the more important the SHAP value is for predicting age.

**Figure 5 brainsci-12-00579-f005:**
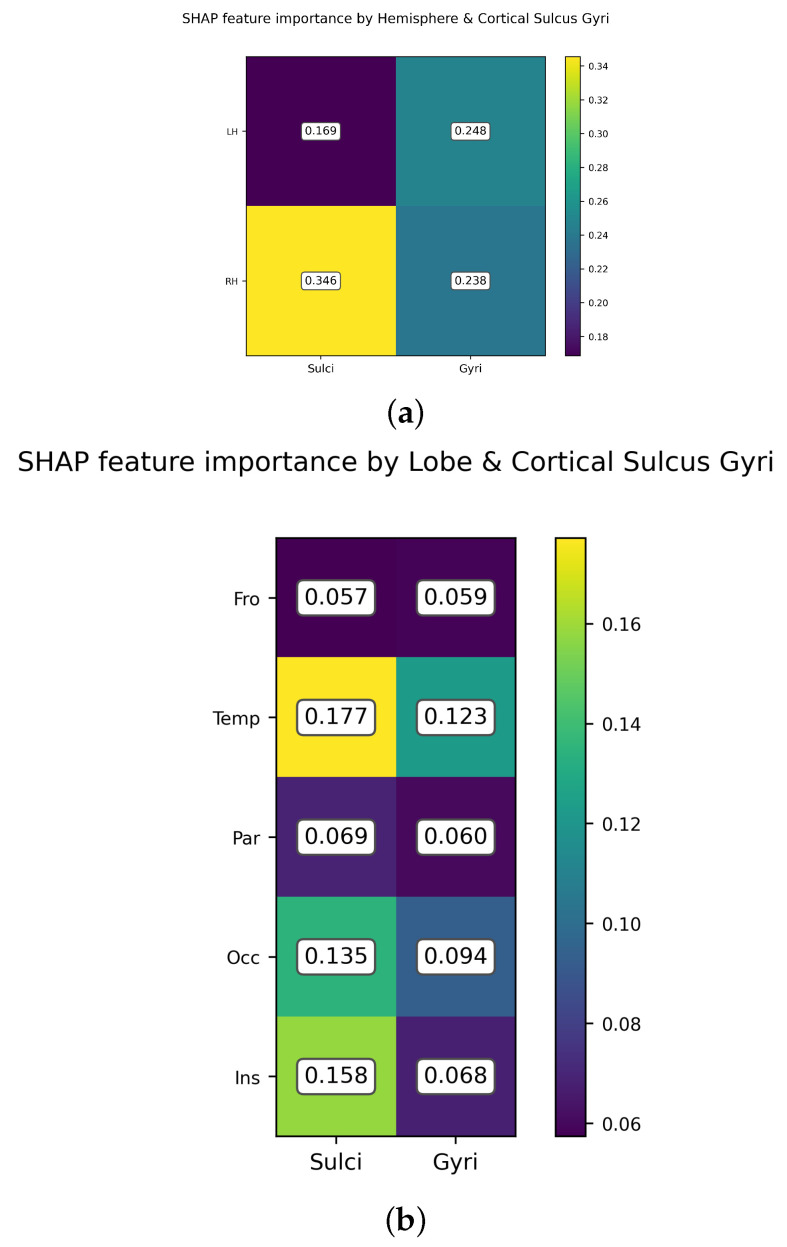
SHAP importance grouping cortical areas by hemisphere, lobe, and type of fold. (**a**) SHAP feature importance in relative terms for brain cortical areas depending on the hemisphere and the cortical surface type (sulci, gyri). As shown in the figure, the aggregate importance of sulci in the right hemisphere for predicting chronological age is 0.346 and is computed as the mean of the SHAP values of right sulci areas normalized by the total of sulci and gyri areas in both hemispheres. (**b**) SHAP feature importance in aggregate for brain cortical areas falling in brain lobes—frontal, occipital, parietal, temporal—and the insula. According to the SHAP values calculated, the temporal lobe contains more information for predicting age than the sulci and gyri-located regions in the other brain lobes.

**Table 1 brainsci-12-00579-t001:** The subjects participating in the study had at least one MRI. After anomaly detection, a total of 3514 neuroradiological assessments were analyzed, with the resulting volumetric analysis shown in Table 2. The table shows the description of the demographic and genetic variables included in the study. M: male, F: female, APOE ϵ(23)ϵ(23): lacking allele ϵ4, ϵ(23)ϵ4 one allele ϵ4 and ϵ4ϵ4 both alleles ϵ4. Educational attainment level, None: no formal education, Primary: primary education degree, Secondary: secondary high school, University: university studies.

SAMPLE SIZE (MRI Scans) = 3514	$N_{\mathrm{SUBJECTS}}=948$
Age	μ=74.68±σ=3.85
Sex	F 519, M 280
APOE	ϵ(23)ϵ(23):145,ϵ(23)ϵ4:647,ϵ4ϵ4: 7
Education	None 150, Primary 232,
	Secondary 203, University 214

**Table 2 brainsci-12-00579-t002:** The table columns from left to right include the name of the subccortical or cortical area, the gray matter volume (mm${}^{3}$) for subcortical structures, the cortical thickness (mm) defined for areas in the surface cortical atlas used, and the *p*-values of the *T*-test and the ANOVA test for Sex and APOE4 variables, respectively. The first row of data shows the brain-to-intracranial-volume (Brain2ICV) ratio, followed by the subcortical structures: thalamus, putamen, amygdala, pallidum, caudate, hippocampus, and accumbens. Next, the parcellation defined in [27], in which, the cortex is divided into gyral and sulcal regions. The brain areas names on the left column are self-descriptive, with LH and RH referring to each hemisphere: the first letter S, G refers to gyral or sulcal thickness. (<0.05 (*) <0.01 (**)).

	Volume (mm${}^{3}$)	Sex p-val	APOE PR (>F)
Brain2ICV	0.698±0.029(%)	**	>0.05
LH Th	6000±653	**	>0.05
RH Th	5848±582	**	>0.05
LH Pu	3551±387	**	>0.05
RH Pu	3635±415	**	>0.05
LH Am	1414±215	**	**
RH Am	1263±193	**	*
LH Pa	1721±252	**	>0.05
RH Pa	1653±244	**	>0.05
LH Ca	3249±490	**	>0.05
RH Ca	3430±252	**	>0.05
LH Hp	3551±387	**	>0.05
RH Hp	3634±415	**	**
LH Ac	462±91	**	>0.05
RH Ac	465±90	**	>0.05
	**Avg.Thickness (mm)**	**Sex (p-val)**	**APOE Pr (>F)**
RH STempSuperior	2.3±0.2	**	>0.05
LH STempSuperior	2.4±0.1	**	>0.05
RH STempSuperior	2.4±0.1	**	>0.05
LH STempInferior	2.5±0.1	**	>0.05
RH STempInferior	2.4±0.1	**	>0.05
LH SOccTempMedandLingual	2.4±0.2	**	>0.05
RH SOccTempMedandLingual	2.4±0.2	**	>0.05
LH SOccTempLat	2.5±0.2	**	>0.05
RH SOccTempLat	2.5±0.2	**	>0.05
RH GTempMid	2.9±0.1	>0.05	>0.05
LH GTempInf	2.9±1.6	>0.05	>0.05
RH GTempInf	2.8±1.7	>0.05	>0.05
LH GTempSup	2.5±0.1	**	>0.05
RH GTempSup	2.5±0.1	**	>0.05
LH GTempSupPlanPolar	3.1±0.2	*	>0.05
RH GTempSupPlanPolar	3.0±0.72	**	>0.05
LH GTempSupLateral	3.0±0.1	**	>0.05
RH GTempSupLateral	2.4±1.7	>0.05	>0.05
LH GTempSupTransv	2.4±1.7	**	>0.05
RH GTempSupTransv	2.5±1.8	**	>0.05
RH SIn	2.1±0.1	**	>0.05
LH SFrontSup	2.1±0.1	**	>0.05
RH SFrontSup	2.1±0.1	**	>0.05
LH SFrontMid	2.1±0.1	**	>0.05
RH SFrontMid	2.1±0.1	>0.05	>0.05
LH SFrontInf	2.2±0.1	>0.05	>0.05
RH SFrontInf	2.1±0.1	>0.05	**
LH SFrontSup	2.2±0.1	**	>0.05
RH SFrontSup	2.1±0.1	**	>0.05
LH GFrontSupp	2.6±1.1	**	>0.05
RH GFrontSupp	2.6±1.1	**	>0.05
LH GFrontMid	2.5±1.1	**	>0.05
RH GFrontMid	2.5±1.1	**	>0.05
LH GFrontInfTriangul	2.5±1.6	**	
RH GFrontInfTriangul	2.5±1.4	**	
LH GFrontInfOrbital	2.7±0.2	**	>0.05
RH GFrontInfOrbital	2.6±0.2	**	>0.05
LH GFrontInfOpercular	2.65±0.13	>0.05	>0.05
RH GFrontInfOpercular	2.65±0.13	*	>0.05
LH GCingPostV	2.4±0.3	*	>0.05
RH GCingPostV	2.5±0.3	>0.05	>0.05
LH SCingMarginalis	2.1±0.1	>0.05	>0.05
RH SCingMarginalis	2.1±0.1	**	>0.05
LH SSubParietal	2.2±0.1	**	>0.05
RH SSubParietal	2.3±0.1	**	>0.05
LH SSubOrbital	2.3±0.2	>0.05	>0.05
RH SSubOrbital	2.3±0.3	*	**
LH SPreCentralSuperior	2.3±0.1	**	>0.05
RH SPreCentralSuperior	2.3±0.1	**	>0.05
LH SPreCentralInferior	2.3±0.1	**	>0.05
RH SPreCentralInferior	2.3±0.1	>0.05	>0.05
LH SPostCentral	2.1±0.1	**	>0.05
RH SPostCentral	2.1±0.1	**	>0.05
RH SPeriCallosal	1.8±0.3	**	>0.05
LH SParietoOcc	2.2±0.1	**	>0.05
RH SParietoOcc	2.2±0.1	**	>0.05
LH SOrbMedOlfact	2.1±1.6	**	>0.05
RH SOrbMedOlfact	2.1±1.5	**	>0.05
LH SOrbitalLat	2.1±1.1	**	>0.05
RH SOrbitalLat	2.1±1.1	**	*
LH SOrbitalHShaped	2.6±1.2	**	>0.05
RH SOrbitalHShaped	2.5±1.2	**	>0.05
LH SOccMideandLunatus	2.3±1.2	**	>0.05
RH SOccMideandLunatus	2.3±1.1	**	>0.05
LH SIntraParietandPariettrans	2.1±0.1	**	
RH SIntraParietandPariettrans	2.1±0.1	**	
RH GParietalSup	2.3±1.2	**	>0.05
LH GParietInfSupramar	2.6±1.3	**	>0.05
RH GParietInfSupramar	2.6±1.2	**	>0.05
LH GParietInfAngular	2.5±1.2	**	>0.05
RH GParietInfAngular	2.5±1.3	**	**
LH SCollatTransvPost	2.1±0.1	**	>0.05
RH SCollTatransvPost	2.1±0.1	**	>0.05
LH SCollTransvAnt	2.6±0.2	**	>0.05
RH SCollTransvAnt	2.5±0.2	**	>0.05
LH PoleOcc	3.3±0.2	>0.05	>0.05
RH PoleOcc	3.3±0.2	**	>0.05
LH GOccSup	2.1±1.1	**	>0.05
RH GOccSup	2.4±1.3	**	>0.05
LH GOccMid	2.5±0.1	**	>0.05
RH GOccMid	2.5±0.1	**	>0.05
LH GOccTempMedParahip	3.1±0.6	**	>0.05
RH GOccTempMedParahip	3.2±0.2	**	>0.05
LH GOccTempMedLingual	2.1±0.1	>0.05	>0.05
RH GOccTempMedLingual	2.1±0.1	>0.05	>0.05
LH GOccTempLatFusi	2.8±1.5	>0.05	>0.05
RH GOccTempLatFusi	2.8±1.5	**	>0.05
LH SInsSup	2.4±0.1	>0.05	>0.05
RH SInsSup	2.4±0.1	**	**
LH SInsInf	2.6±0.2	**	>0.05
RH SInsInf	2.5±0.1	>0.05	>0.05
LH SCircInsAnt	2.7±0.2	>0.05	>0.05
RH SCircInsAnt	2.7±0.2	**	>0.05
LH GInsularShort	3.4±0.3	**	>0.05
RH GInsularShort	3.4±0.3	**	>0.05
LH GCentInsula	3.2±0.3	**	>0.05
RH GCentInsula	3.3±0.3	**	>0.05
LH SCentral	2.0±0.1	**	>0.05
RH SCentral	1.9±0.1	**	>0.05
LH GPreCentral	2.6±1.7	**	>0.05
RH GPreCentral	2.6±1.6	**	>0.05
LH GPostCentral	2.1±1.5	**	>0.05
RH GPostCentral	2.1±1.5	**	*
LH SCalcarine	1.9±0.1	>0.05	>0.05
RH SCalcarine	1.9±0.1	**	>0.05
LH GRectus	2.5±0.2	>0.05	>0.05
RH GRectus	2.5±0.2	>0.05	>0.05
LH GOrbital	2.7±1.7	**	>0.05
RH GOrbital	2.7±1.7	**	>0.05
LH GCuneus	1.9±0.1	>0.05	>0.05
RH GCuneus	1.9±0.1	**	>0.05
LH LatFisPost	2.3±0.1	**	>0.05
RH LatFisPost	2.3±0.1	**	*
LH LatFisAntHoriz	2.1±0.1	>0.05	>0.05
RH LatFisAntHoriz	2.4±0.1	**	>0.05

**Table 3 brainsci-12-00579-t003:** Performance metric of PLS and XGBoost models in the test set (unseen subjects). The performance measures shown in the table are the maximum absolute error (MAE), the maximum residual error (MXE), the mean absolute percentage error (MAPE), and the median absolute error (MEDAE).

	Test Performance Measure			
Model	MAE	MXE	MAPE	MEDAE
PLS	2.570177	10.2293404	0.03348523	2.15809710
XGBoost	2.0301	8.7138485	0.0265	1.74578

## Data Availability

Code and data used in this research are publicly available on the Github repository under an Apache 2.0 license at https://github.com/grjd/chronological_brain_age, accessed on 1 April 2022.

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
