# Peer review of "Prediction of Chronological Age in Healthy Elderly Subjects with Machine Learning from MRI Brain Segmentation and Cortical Parcellation"

_brainsci, 2022, doi:10.3390/brainsci12050579_

Round 1

Reviewer 1 Report

1-The authors discuss why they used XGBoost algorithm rather than other prediction models. Also, a recent study has shown that the type of prediction algorithm has significant impact on brain age values. This matter should be discussed in the paper.

Predicting brain age using machine learning algorithms: A comprehensive evaluation

2- The main contribution of this study is not clear in the introduction. What is the gap knowledge which this study is going to address it.

3-I would suggest to open a subsection about other studies and comprehensively discuss about them in that section.

4-Study Participants should be detailed. It would be better to summarize into a table.The number of participants is not match in the abstract and main text.

6-The big table in page 10 and Fig 2 can be moved to the supplementary information.

7- Table 4, other algorithms such as SVR should be evaluated on this dataset.

8-How about bias adjustment on the results? The impact of bias-adjustment should be discussed.

Bias-adjustment in neuroimaging-based brain age frameworks: A robust scheme

9-What is the color bar in Fig .5?

10-The authors should open a “discussion” section and Interpreter the achieved results and compare them with other studies.

Author Response

1-The authors discuss why they used XGBoost algorithm rather than other prediction models. Also, a recent study has shown that the type of prediction algorithm has significant impact on brain age values. This matter should be discussed in the paper.

A Discussion section has been included now in the manuscript, there the comprehensive evaluation in \cite{beheshti2021predicting} and related studies in machine learning methods for age prediction are discussed. The added text in the Discussion text is shown below for the reviewer’s convenience.

            “For a comprehensive evaluation of machine learning algorithms for age prediction see \cite{beheshti2021predicting}. Support Vector Regression (SVR) achieved the best accuracy in predicting age in healthy subjects, MAE $\sim$ 3 and MAE $\sim$ 5 years for training and test sets (N=788+88). The performance of the SVR in our dataset is shown in the Supplementary Material. The reduction in the prediction error shown in our study may respond to several factors such as the large single-center dataset, the scalability of the tree boosting algorithm, and the mapping from input features to output data. The input space in this study is the product of segmentation and parcellation methods to extract subcortical volumes and cortical thickness which is admittedly more computationally demanding than grey matter intensities extracted from whole-brain \cite{cole2015prediction}, \cite{wang2019gray}.”

2- The main contribution of this study is not clear in the introduction. What is the gap knowledge which this study is going to address it.

           The study aims at investigating the effect of age on global and regional brain volumes and cortical thickness. This is stated in the Abstract and the last and the antepenultimate paragraphs of the Introduction. Furthermore, the study tries to deliver insights to shorten the explanatory gap between chronological age and biological brain age. As discussed both in the Introduction and Conclusion sections, our model predicts chronological age and not biological brain age which cannot be directly measured as is the case with the chronological age and can, however, be estimated via proxies such as DNA damage.

3- I would suggest to open a subsection about other studies and comprehensively discuss about them in that section.

            A new Discussion section has been included, wherein the implications and limitations of the results described in the manuscript are reviewed concerning relevant studies. The added text in the Discussion text is shown below for the reviewer’s convenience.

“While this study benefits from being a single-center using an identical protocol for image acquisition, the generalization of prediction models coming from multi-site datasets is becoming a crucial problem in need of a solution\cite{smith2019harmonizing}. A promising approach to achieving acceptable harmonization and coherence prediction results is transfer learning \cite{valverde2021transfer}. A recent study \cite{chen2020generalization} used the transfer learning approach to build brain age prediction models from diffusion MRI datasets extracted from the CamCAN repository \cite{shafto2014cambridge} and achieving a MAE within 4 to 5 years. Although not directly comparable, it is worth noting that the range of chronological ages in our study is within 69-88 years, while Chen and colleagues try to predict from the adult population (18 and older) with a smaller dataset (616 samples).

For a comprehensive evaluation of machine learning algorithms for age prediction see \cite{beheshti2021predicting}. Support Vector Regression (SVR) achieved the best accuracy in predicting age in healthy subjects, MAE $\sim$ 3 and MAE $\sim$ 5 years for training and test sets (N=788+88). The performance of the SVR in our dataset is shown in the Supplementary Material. The reduction in the prediction error indicated in our study may respond to several factors such as the large single-center dataset, the scalability of the tree boosting algorithm, and the mapping from input features to output data. The input space in this study is the product of segmentation and parcellation methods to extract subcortical volumes and cortical thickness which is admittedly more computationally demanding than grey matter intensities extracted from whole-brain \cite{cole2015prediction}, \cite{wang2019gray}.”

4-Study Participants should be detailed. It would be better to summarize into a table.The number of participants is not match in the abstract and main text.

The Study Participants subsection of the Methods Section has been expanded and corrected in the new version of the manuscript. Table 1 now includes the Sample size, this was missing in the previous version of the manuscript as correctly pointed out by the reviewer.

“The MRI scans were performed in a time span of 6 years. The number of MRI examinations that passed visual inspection and automatic anomaly detection is as follows: year 1 857 scans, year 2 683 scans, year 3 658 scans, year 4 555 scans, year 5 450 scans, year 6 311 scans for a total of 3514 MRI examinations.” 

Furthermore, a more thorough description of the cohort is now included in the Methods Section with a description of the methodology used for the cognitive assessment of the subjects in the study.

“Of the initial 1213 subjects, those diagnosed with mild cognitive impairment (MCI) or dementia were excluded, resulting in a cohort of 948 healthy elderly subjects. Cognitive status was determined with the Mini-Mental Status Examination (MMSE), Free and Cued Selective Reminding Test (FCSRT), Semantic fluency, 101 Digit-Symbol Test and Functional Activities Questionnaire (FAQ).”

6-The big table in page 10 and Fig 2 can be moved to the supplementary information.

            The table containing the volumetric, thickness, and the statistical tests is exceedingly large but this is a direct consequence of the comprehensive and exhaustive volumetric and cortical analysis performed in the study, which uses the cortical Destrieaux Atlas. Admittedly both the Table and Figure 2 are large and may interrupt the reading flow of the text. We are open to the reviewer’s suggestion in accordance with the formatting choice of the journal. The enumeration of the Table has been corrected to Table 2.   

7- Table 4, other algorithms such as SVR should be evaluated on this dataset.

                 The results obtained via the implementation of Support Vector Machine with hyperparameter tunning using Grid Search Cross-Validation are shown in the Supplementary Material. SVR shows performance comparable to XGBoost. There are, however, multiple algorithmic choices, nevertheless, in this study, we aim at gaining an understanding of the most important features (cortical thickness) to predict chronological age. There is abundant literature establishing the capacity of machine learning algorithms such as support vector machine and ensemble learning methods among others to predict brain age, and a discussion of some relevant studies have been highlighted in the manuscript. However, this study aims not only to build predictive models but most importantly to explain which are the most important features via SHAP values.    

8-How about bias adjustment on the results? The impact of bias- adjustment should be discussed.

This important suggestion has been incorporated in the Introductory section, providing a discussion of the rationale for reducing the delta or difference between the estimated brain age and the chronological age. The improvement in robustness with the bias adjustment technique is discussed together with upcoming molecular metrics for cellular senescence which together may help shorten the explanatory gap between biological and chronological age. Below is shown the updated text in the Introduction Section for the reviewer’s convenience.

"Brain age" is used in the machine learning literature as the age estimate in a regression model \cite{cole2018brain}. Accordingly, the delta or difference between the brain age (estimated) and the chronological age (given) would be indicative of the health status of the subject. However, for this approach to be sensible, it requires a model producing accurate estimates of the biological age.  In \cite{beheshti2019bias} a bias-adjustment technique was shown to shorten the delta or difference between chronological and brain estimated age. However, the pertinence of any approach that tries to reduce the above-mentioned delta, will need to be confronted with direct assessments of brain age. Recently, new cellular and molecular approaches for brain senescence and decline have been put forward. For instance, transcriptome profiling \cite{ham2020advances}, DNA methylation \cite{hernandez2011distinct}, \cite{horvath2012aging}, \cite{hannum2013genome} or immune metrics such as the inflammatory clock of aging (iAge) \cite{sayed2021inflammatory} aim at capturing cellular senescence holding promise for understanding longevity and neurodegenerative processes.”

9-What is the color bar in Fig .5?           

     We thank the reviewer for making this point and allowing us to properly describe the metric used in Figure 5. All the features included in the cortical atlas are grouped according to their hemisphere and surface pattern (girus or sulcus) in Figure 5 (a) and according to surface pattern and brain lobe (Frontal, Temporal, Parietal, Occipital, Insula) in Figure 5 (b) . For each subgroup, we compute the aggregate SHAP feature importance as the sum of the SHAP value for each region. The color bar represents the relative importance of each subgroup in relation to the other groups considered according to the SHAP values. For example, Figure 5 a) shows that sulci type parcellations in the right hemisphere are more important (in the SHAP value sense) than sulci in the opposite hemisphere. The last paragraph on Page 17 contains a description of the specifics and is shown below for the reviewer’s convenience.

“Figure 5 depicts the aggregate importance of sulci and gyri cortical areas according to hemispheres 5a and lobes 5b. The aggregate importance is computed as the mean average of the SHAP values normalized. According to the SHAP values, the total sulcus thickness in the right hemisphere contains on average more information regarding the chronological age of the subject than gyrus thickness in either hemisphere (0.238, 0.248) and more than sulcus thickness in the left hemisphere (0.169).”

10-The authors should open a “discussion” section and Interpreter the achieved results and compare them with other studies.

A new Discussion Section has been included, distinguishing now as suggested by the reviewer, between Results, Discussion, and Conclusions. The implications of the results described in the manuscript are presented in comparison with related studies with the references herein.

Reviewer 2 Report

Summary: The authors developed a prediction model to predict chronological age using brain MRI features. The methods and the findings of the study are very interesting. Details involving motivation of the study, methods and results were clearly explained.

Review Comments:
1. Overall the manuscript looks good. Though the authors need to work on structuring the manuscript, reorganizing the content to appropriate sections. For example, lot of intermediate results were discussed in methods section which has to be moved to supplementary or results section.
Results section talks about methodological details which has to be restructured to methods section.
2. Results and Discussion are written together which looks confusing for the readers. the authors need to follow the research manuscript standard writing to distinguish clearly between these sections
3. The population includes a longitudinal data but I do not see longitudinal analysis carried out in this study, how did the authors handle the data from same subjects at different time points, are they simply included as separate samples in the prediction model. how is it taken care of?
4. Add references for "generating an isotropic brain image with non-brain tissue removed"
5. Table 1: Subject demographics should include break up of the sample size as per different time points
6. While the findings of XGBoost were compared with conventional PLS method, which are very much expected results. Comparing the method with state of the art methods published related to age prediction (eg., Chen2020)  would add significance and is of interest to the readers.

Chen2020 Chen, Chang-Le, Yung-Chin Hsu, Li-Ying Yang, Yu-Hung Tung, Wen-Bin Luo, Chih-Min Liu, Tzung-Jeng Hwang, Hai-Gwo Hwu, and Wen-Yih Isaac Tseng. "Generalization of diffusion magnetic resonance imaging–based brain age prediction model through transfer learning." NeuroImage 217 (2020): 116831.

Author Response

1. Overall the manuscript looks good. Though the authors need to work on structuring the manuscript, reorganizing the content to appropriate sections. For example, lot of intermediate results were discussed in methods section which has to be moved to supplementary or results section.
Results section talks about methodological details which has to be restructured to methods section.

We thank the reviewer for this suggestion, the results of the statistical test were indeed in the previous version included in the Methods section. The results of the hypothesis testing to investigate the effect of sex and APOE4 are now described in the Results section. 

Table 1 now included the sample size. We describe the number of MRI scans performed each year and the statistics on the number of scans per subject.

The first paragraph in Methods.Statistical Analysis section has been extended with the below text

 “The MRI scans were performed in a time span of 6 years. The number of MRI examinations that passed visual inspection and automatic anomaly detection is as follows: year 1 857 scans, year 2 683 scans, year 3 658 scans, year 4 555 scans, year 5 450 scans, year 6 311 scans for a total of 3514 MRI examinations.” 

 Methods.Study participants section includes now a description of the methodology used for the cognitive assessment of the subjects in the study and is shown next for the reviewer’s convenience. 

"Of the initial 1213 subjects, those diagnosed with mild cognitive impairment (MCI) or dementia were excluded, resulting in a cohort of 948 healthy elderly subjects. Cognitive status was determined with the Mini-Mental Status Examination (MMSE), Free and Cued Selective Reminding Test (FCSRT), Semantic fluency, 101 Digit-Symbol Test and Functional Activities Questionnaire (FAQ)."

Subsection II.ii is now included within section II.i “Study Participants” including the ethical approval code 

2. Results and Discussion are written together which looks confusing for the readers. the authors need to follow the research manuscript standard writing to distinguish clearly between these sections

    A new section Discussion has been included, distinguishing now as suggested by the reviewer between Results, Discussion, and Conclusions.

3. The population includes a longitudinal data but I do not see longitudinal analysis carried out in this study, how did the authors handle the data from same subjects at different time points, are they simply included as separate samples in the prediction model. how is it taken care of?

The dataset originates from a study a community-based longitudinal study addressing the early detection of AD. The participants underwent a detailed assessment protocol including annual neurological and neuropsychological examinations. The study has brought forth the application of different methodologies to neurodegeneration and aging, including a recent study transition from mild cognitive impairment to normal condition exploiting the temporal dimension of the study \cite{gomez2020selecting}, \cite{gomez2021comparative}, \cite{sanz2021transition}. Here we are interested in age-related brain changes observed in brain volumetric analysis and cortical parcellation. We perform a (chronological) age-based analysis of cortical and volumetric atrophy rather than an individual-based examination of personal atrophy. The subjects in the study paid a different number of visits (between 1 to 6). As properly identified by the reviewer the training dataset includes all the samples irrespective of the subject identifier, the actual visit number of the subject is immaterial to predict the chronological age of the subjects. 

Nonetheless, we include now in the Methods. Study participants a description of the temporal provenance (MRI scan performed in year 1 to year 6) of the totality of samples used in this study.

The distribution of the number of visits with the corresponding neuroradiological and cognitive assessment is as follows: $19.2\%$ of subjects with one initial assessment completed, $11.3\%$ of subjects with 2 assessments completed, $8.5\%$ with 3 assessments completed, $8.9\%$ of subjects with 4 assessments completed, $23$ of subjects with 5 assessments and $29.1\%$ of subjects with all 6 assessments completed.

4. Add references for "generating an isotropic brain image with non-brain tissue removed"

The Reference mccarthy2015comparison has been included in the text, section "Methods. MRI Data Acquisition and Preprocessing".

5. Table 1: Subject demographics should include break up of the sample size as per different time points

We thank the reviewer for making this point which helps to recognize the sample size. Table 1 originally indicated only the number of subjects participating in the study, however, the sample size as the reviewer correctly suggests is the number of MRI scans that were performed since all subjects have one or more MRI scans. The sample size is now properly highlighted in Table 1 (3514) with an additional description of the year in which the scans were collected. As already mentioned in point 3, the distribution of the number of visits with the corresponding neuroradiological and cognitive assessment can now be found in the Methods. Study participants section.

6. While the findings of XGBoost were compared with conventional PLS method, which are very much expected results. Comparing the method with state of the art methods published related to age prediction (eg., Chen2020) would add significance and is of interest to the readers.Chen2020 Chen, Chang-Le, Yung-Chin Hsu, Li-Ying Yang, Yu- Hung Tung, Wen-Bin Luo, Chih-Min Liu, Tzung-Jeng Hwang, Hai- Gwo Hwu, and Wen-Yih Isaac Tseng. "Generalization of diffusion magnetic resonance imaging–based brain age prediction model through transfer learning." NeuroImage 217 (2020): 116831 

A new Discussion section has been included, wherein the implications and limitations of the results described in the manuscript are reviewed concerning relevant studies. The added text in the Discussion text is shown below for the reviewer’s convenience.

"While this study benefits from being single-center using an identical protocol for image acquisition, the generalization of prediction models coming from multi-site datasets is becoming a crucial problem in need of a solution\cite{smith2019harmonizing}. A promising approach to achieving acceptable harmonization and coherence prediction results is transfer learning \cite{valverde2021transfer}. A recent study \cite{chen2020generalization} used transfer learning approach to build brain age prediction models from diffusion MRI datasets extracted from the CamCAN repository \cite{shafto2014cambridge} and achieved  MAE within 4 to 5 years. Although not directly comparable, it is worth noting that the range of chronological ages in our study is within 69-88 years, while Chen and colleagues try to predict from an adult population (18 and older) with a smaller dataset (616 samples)."

Round 2

Reviewer 1 Report

The authors successfully addressed to my concerns. However, there are some comments which authors should consider them in this study.

  • The authors should carefully check all abbreviations. For example, the abbreviation “MRI” has not been used in abstract.
  • The author should provide more details about SHAP strategy rather than only referring to other studies.
  • Which version of Freesurfer was used in pre-processing stage (i.e., 6 or 7)?
  • Define the variables in the following equation:

Y = b0 + b1XSex.

  • Have the authors normalized (to remove the effect of head size) the variables in Table 3 for sex differences? Or the statistical results have been achieved based on raw measurements?
  • Define all variables (e.g., MAE, …) used in table 4 as note.
  • Both prediction models showed a very good performance in terms of MAE, but R2 is mediocre. Any reason for that?
  • It would be nice if the authors plot the predicted age vs. real age for both training and hold-out sets.
  • In figure 4, what is “free”? for example, free_R_Hipp.

Author Response

Reviewer 1 Second Round

The authors successfully addressed to my concerns. However, there are some comments which authors should consider them in this study.

The authors should carefully check all abbreviations. For example, the abbreviation “MRI” has not been used in abstract.

We went through all the abbreviations and acronyms used in the manuscript and corrected in the Results Section the line in which the abbreviations PLS and XGBoost acronyms were used before they were defined, correcting when necessary as for example:

“shows the results for the age predictors built, linear (PLS) and nonlinear (XGBoost).”

“shows the results for the age predictors built, linear and nonlinear.”

The MRI term in the abstract is intended to define the acronym, rather than as an abbreviation per se, however, the acronym is surely well known to the audience of Brain Sciences and therefore we agree with the reviewer in deleting MRI from the abstract. The acronym MRI is defined in the Introduction section.

The author should provide more details about SHAP strategy rather than only referring to other studies.

            We agree with the reviewer that the manuscript would benefit with a more complete description of the SHAP values approach, always within the limitations of scope and space. A comprehensive definition of Shapley value has been included in the Methods section and shown next for the reviewer’s convenience:

Formally, the Shapley value Φ is defined via a value function ν of all features in a set S. Specifically, the Shapley value of a feature value is its contribution to the payout (e.g. if the average prediction for all instances is 0.9 and the actual prediction is 0.8, the payout of 0.1) weighted and summed over all possible value combinations.

where p is the number of features, S is the subset of features, x is the vector of feature values of the particular instance to be explained and ν(S) is the prediction for feature values in S, marginalized over features that are not included in the set S.

The Shapley value is arguably the best permutation-based method for ex- plaining the effects of feature values in the average prediction. An important drawback of Shapley values is that they provide additive contributions (attribu- tions) of explanatory variables. If the model is not additive, then the Shapley values may be misleading. For a more in detail description of SHAP values see [Gómez-Ramírez et al., 2020a] and references within. “

Which version of Freesurfer was used in pre-processing stage (i.e., 6 or 7)?

            The Freesurfer version used in the pre-processing stage is version 6 release date 2019-03-28

Define the variables in the following equation: Y = b0 + b1XSex.

            The definition of the variables used in the model is now included in the text:

obtained in the regression model Y = β0 + β1XSex where the target variable Y is the chronological age and XSex is the sex of the participants.

Have the authors normalized (to remove the effect of head size) the variables in Table 3 for sex differences? Or the statistical results have been achieved based on raw measurements?

            We thank the reviewer for this remark since it allows us to give important details about brain segmentation that were missing in the previous version. The following text has been added in The Methods. Statistical analysis section:

“The total intracranial volume (eTIV) estimated by FreeSurfer has previously been reported to have linear correlations of 0.9 with manually estimated intracranial volume [Shen et al., 2010], [Malone et al., 2015]. Depending on whether CSF is included or not, one can dissociate the total brain volume (TBV) from the total intracranial volume. The normalized TBV is widely used as an index for brain atrophy, as the head size remains stable across the life span and serves as a good measure to reduce between-subject differences with regard to maximum brain size. Whole-brain volume, on the other hand, changes throughout the life span of an individual. Measurements of total brain volume (TBV) with FreeSurfer are robust across field strength [Heinen et al., 2016].

The total intracranial volume acts as a scaffolding of the brain and sets an upper bound for the brain’s volume. Accordingly, it is possible to build a proxy of the brain atrophy that an elder person went through her adult life by computing the ratio between the brain volume (TBV) and the total intracranial volume (eTIV) which represents the upper limit of brain volume. Thus, Brain2ICV = TBV/eTIV .”

Define all variables (e.g., MAE, ...) used in table 4 as note.

            The measures used in the table are now defined in the Table’s caption and are shown next for the reviewer’s convenience.

Table 4: Performance metric of PLS and XGBoost models in the test set (unseen subjects). The performance measures shown in the table are the maximum absolute error (MAE), the maximum residual error (MXE), the mean absolute percentage error (MAPE), and the median absolute error (MEDAE).

Both prediction models showed a very good performance in terms of MAE, but R2 is mediocre. Any reason for that?        

The R2 is the coefficient of determination score and is used to evaluate the performance of linear regression models but is suboptimal for nonlinear models and should not be trusted (e.g. https://pubmed.ncbi.nlm.nih.gov/20529254/). R2 is not appropriate to evaluate goodness of fit of simulated or predicted (Y’) versus observed (Y). Importantly, the coefficient of determination R2 quantifies the degree of any linear correlation between observed and predicted values, while for the goodness-of-fit evaluation only one specific linear correlation should be taken into consideration (the 45 degree line). The table provides several goodness of fit measures giving an overall picture of the algorithms' performance with the caveat that the interpretation of the measures should be paired with the mathematical framework of the model. The R2 is being removed to favor measures that portray direct comparability and interpretation across models. However, R2 remains in the text in the proper context of a comparison of the algorithms with the dummy models, for example a constant model that always predict input features will have a R2 =0.0 (the R2 score in the implementation here used can also be negative).  

It would be nice if the authors plot the predicted age vs. real age for both training and hold-out sets.

            A scatter plot displaying the actual values along the X axis and the predicted values along the Y axis is provided in Figure 1 of Supplementary Material

In figure 4, what is “free”? for example, free_R_Hipp.

            The “free” and “fr” are abbreviations of Freesurfer and have been deleted from the Figures for the sake of clarity.
